# Everyday Life Construction, Outdoor Activity and Health Practice among Urban Empty Nesters and Their Companion Dogs in Guangzhou, China

**DOI:** 10.3390/ijerph17114091

**Published:** 2020-06-08

**Authors:** Xianfei Chen, Hong Zhu, Duo Yin

**Affiliations:** 1School of Geography, South China Normal University, Guangzhou 510631, China; chenxianfei@m.scnu.edu.cn; 2School of Geographical Sciences, Guangzhou University, Higher Education Mega Centre, Guangzhou 510006, China; zhuhong@gzhu.edu.cn

**Keywords:** companion dog, urban empty nesters, multispecies kinships, outdoor activities, health practice

## Abstract

In this paper, we argue that research on the everyday life of older people needs to move beyond anthropocentrism because non-human support contributes to the diversity of their social networks. We elaborate this argument by examining how companion dogs are involved in the urban empty-nest family in Guangzhou (an aging and highly urbanized city in China), the building of multispecies kinships by urban empty nesters in later life and improving the health of urban empty nesters. Participatory observations and 20 in-depth interviews were combined to assess the association between dog ownership and the reconstruction of later life. Specifically, we focus on the co-disciplined pursuit of outdoor activities by urban empty nesters and their companion dogs; this pursuit represents a shared leisure practice that maintains multispecies kinship and is a creative way for older individuals to improve their happiness and physical functioning. This paper provides a relational and reflective understanding of the interaction between the urban empty nesters and companion dogs and the implications of this interaction in the urban leisure space.

## 1. Introduction

In recent years, the Chinese government has promoted a more active lifestyle for seniors to improve their health and well-being. This discourse encourages older people to spend their remaining years positively; stay healthy and independent; and thereby delay physical decline, prevent diseases, and improve their quality of life. One’s later years are thus considered a period of enjoyment, challenge, growth, and exploration [1]. China has begun to age and is faced with such major issues as low fertility, an aging population, and an inadequate social security system, all obstructing the country’s path of development. In addition, China is experiencing accelerating population mobility and rapid urbanization. New discursive constructions tend to link “old age” to the changing demographic situation and the reconstitution of urban households [2]. In this situation, urban empty nesters are more likely than those living with their children to face such difficulties as lack of social support (especially from family members) and limited urban space. Current research on old age in urban China has focused primarily on the quality of intergenerational relations [3], the social support available to older people [4], and community services for older people [5]. Our aim is quite different; we will utilize a relational angle about leisure and multispecies interaction [6] to delve deeply into the life of urban empty nesters.

The term “empty nest” usually refers to the years a couple spends together between the departure of their last child from the home and the death of one of the spouses [7]. Empty nesters usually include empty-nest couples and empty-nest individuals. Smaller family sizes, the closer spacing of children, and increased longevity have resulted in longer empty-nest periods. Empty nests will grow in number with the world’s aging population [8]. For most older people in Western countries, the empty-nest period, which Glenn [9] refers to as the “postparental stage”, begins after the children graduate from high school or reach 18 years old. In contrast, in China, under the influence of generational support and traditional filial piety, the vast majority of the older people tend to live with their children, receive intergenerational support from family members, and take the initiative to look after their grandchildren. Several generations living under one roof is common in traditional Chinese households. Historically, older people were proud to have a large group of children and grandchildren surrounding them [2].

However, as the development of the economy, society and culture have led to economic surplus and differences in intergenerational ideas, a growing number of young people are leaving their parents and prefer to hire professional babysitters. As a result, an increasing number of older people have begun to live alone as empty nesters without the company of their children and have become vulnerable to severely insufficient social support. China has the world’s largest aged population, and empty nesters represent 25% of this group [10]. This proportion is expected to reach 90% by 2030, when almost all senior households will be empty nests [11]. Confined to a small range of activity and social interaction, empty nesters have very limited access to social support [12]. Previous studies have shown that the lack of a strong social support system can negatively affect the physical and mental health of older people by reducing their positive emotions and experiences [13]. As Boermel [2] has observed, the general concern for these “empty nesters” can be interpreted as a social commentary on Chinese cities in the post-reform period. Owing to the rapid growth of China’s economy and the uneven economic development between the eastern and western regions, empty nesters living in rural and urban areas differ significantly from each other in terms of the living environment, education level, income status, and health status [10]. Researchers usually pay more attention to rural empty nesters (who are generally disadvantaged with respect to the above life quality factors) and less attention to the profound impact of rapid urbanization and the aging of cities on older urban dwellers. In contrast, this paper concentrates on the daily life of urban empty nesters in China and places their health in the context of rapid urbanization and aging in contemporary China.

In the past 40 years, social attitudes toward dog ownership in China have shifted significantly. During the pre-reform period (1949–1978), raising pet dogs was condemned as an unpopular bourgeois way of life that had no place in revolutionary China. The vast majority of Chinese people regard them as working dogs and believe that dogs can guarantee personal and property security. In the post-reform era, with the economic transformation, the improvement of living standards, and the end of ideological dogma, these factors have driven the revival of pet-keeping culture [14]. Since the early 1990s, pet ownership has become increasingly popular especially in developed and coastal cities. For example, there are more than 1 million pet dogs registered in Beijing [15], and most of the owners are the wealthy and the elites. In the new century, more and more dogs have entered ordinary families, the companionship of dogs has been valued instead of the entertainment experience it brings to owners. Urban families with only one child and empty nester often choose to keep dogs as companions. According to the *Chinese Pet Industry Development Report 2018*, nearly 6% of households in China own pets [16]. Though this number is much lower than that in the developed world, this number is now growing faster than ever. Elderly urban dwellers are more likely to own a companion dog, as they pay more attention to leisure and have more spare time and money than those living in rural areas. In addition, urban empty nesters often choose to raise companion dogs for companionship due to the lack of necessary support from family members, and urban empty nesters do not have to look after grandchildren. We focus on Guangzhou (a city characterized by a high level of urbanization and a large aging population), which offers a typical snapshot of urban development in China. In this paper, case studies are conducted on empty nesters in Guangzhou from multispecies leisure perspectives. We investigate how empty nesters accept companion dogs into their lives and how multispecies kinship is developed in this process.

Leisure scholars have tended to hold that to acquire a positive mindset in one’s later years, one needs to be free of diseases and disease-related disability, have high cognitive and physical functioning potential, and take an active part in life [17,18,19]. Therefore, having close social connections and engaging in meaningful activities are highly important. How to improve the quality of later life and encourage more older people to become involved in daily leisure has unsurprisingly attracted widespread interest from scholars. For example, leisure activities, such as square dances and opera performances, which are currently very popular among older people in China, represent an active means to improve later life [20,21] and gain effective interpersonal support and group identity [22]. This effort to gain interpersonal support and group identity is reflected mainly in the pleasure of social relations (including meeting new friends and engaging in conversations before and after leisure activities), which tends to emphasize the pleasurable aspects of interpersonal support in leisure interaction. However, leisure practices and spaces for humans often involve multiple species, and non-human animals usually create the experience and meaning of life together with humans [23,24]. The involvement of non-human animals in human leisure practices and spaces pertains to the discussion on the concept of “multispecies leisure” in the landscape of leisure in recent years; this discussion puts the interaction between humans and animals in the framework of leisure, emphasizes the intimate and active leisure relationship between animals and humans [6], and emphasizes the initiative of animals rather than just their attachment to each other. It would be interesting to use this concept as a basis for discussing the non-human support provided by companion dogs for urban empty nesters. Therefore, the multispecies kinship in this paper refers specifically to the family formed by urban empty nesters and their companion dogs.

Numerous anthrozoological and medical studies have revealed that pet ownership also leads to improved mental health [25,26,27]. For instance, dogs can reduce the social isolation of older persons with a high risk for physical illness and emotional disorder by providing companionship [23]. Moreover, as Thorpe et al. [28] note, older people tend to maintain their physical activity longer if they own a dog. In terms of physical health outcomes, researchers found that dogs can help seniors recover from dementia and cognitive decline [29]. In the opinion of Charnetski et al. [30], owner-pet interactions can spur the owner’s body to produce specific neurochemicals that strengthen the immune system. Wells [31] powerfully advocates that such physical benefits are actually mutual; in other words, when owners touch their dogs, the blood pressure of both decreases. In terms of mental health, dogs help to alleviate anxiety, depression, and loneliness for their owners and offer them support and comfort, thus increasing the owners’ sense of well-being [32]. According to another study, for older people, emotional support from pets is more effective than human support [33]. In sum, the above empirical studies tellingly illustrate that the benefits of older people interacting with dogs do exist. Nevertheless, how older people interact with companion dogs for beneficial purposes and why pet ownership helps older people rebuild their lives are poorly understood. This study will, therefore, explore how companion dogs are involved in reconstructing the everyday life of urban empty nesters and how can they achieve active aging of the elderly through leisure interaction?

We argue that the non-human support provided by companion dogs is an effective way to extend social support to urban empty-nesters because companion dogs help their owners reconstruct their everyday lives by forming multispecies relationships with their owners and improve the health of the elderly through leisure interactions. In a word, living a fulfilling later life is associated with positive well-being and ageing. A creative approach, such as encouraging companion dog ownership as part of the effort to improve urban empty nester’s health status through a more diversified social support network, is needed.

## 2. Materials and Methods

This study investigated leisure interactions between urban empty nesters and their companion dogs in Guangzhou, a city located on the southeast coast of China. According to the pension department of the Guangzhou Civil Affairs Bureau, by the end of 2018, the registered older population in Guangzhou had reached 1.693 million, thus accounting for 18.25% of the total population [34]. It is expected that by 2020, the number of older people in Guangzhou will reach 1.83 million, thereby making the city a moderately aging society within an aging population structure that is characterized by the overlap of advanced aging, empty-nest families, and core families. I chose to study the Huashida Community (H community) of Guangzhou for two reasons: First, the H community is located in central Guangzhou and is one of the typical aging communities in the city. Second, the H community has implemented a program of caring for empty nesters and sends them holiday gifts and blessings each year to convey a sense of support and community to them. As a model for elderly care and social harmony in urban communities, the H community is passionately extolled by the local government as a “*Model for Respecting the Elderly*”. Moreover, a long-term service called “*Empty Nesters Care*” is carried out by a group of volunteers in the H community [35]. Lacking strong support from family and friends, empty nesters often choose to keep a dog for company. These empty nesters walk their dogs in the public space of the H community.

In this study, I collected data through participant observation and semi-structured interviews. A two-week observation was conducted in the morning and evening before the formal interview. I observed the time and space of the activities pursued by older people and their companion dogs and the interaction between the dogs and passers-by and found some older people walking their dogs very regularly on the same path. I invited these people to be my interviewees, who provided full informed consent after I explained the purpose of the study.

With the consent of the owner, the typical dog-walking scene, and the owner’s interaction with his or her companion dog was recorded and photographed. Since the research is based on the everyday life experience of urban empty nesters, to understand and reflect on their interaction with companion dogs, I asked if my participation in their walking activities would be convenient for them. This participation allowed the interaction between the owner and the companion dog to be documented most directly, and in-depth interviews were conducted during the dog walking. Follow-up surveys were performed from 10 September to 27 October 2019, and several ongoing internet interviews were performed afterward. All interviews were conducted by the first author.

The respondents, who were all urban empty nesters, were coded No. 1–No. 10 (Table 1) based on the interview order. Interviewee No. 1 is an empty nest couple. Interviewee No. 4 is a family that consists of a mother and her daughter who do not live together. The daughter visited her mother once a month. I happened to meet the daughter during her visit to her mother when the mother agreed to participate in my project and accepted my interview. Interviewee No. 9 is an experienced caregiver. The duration of each interview was 30–90 min, depending on the dog’s daily activity. All interviews were recorded with the consent of the owner and transcribed later. WeChat (the most popular social messaging app in China) was used to facilitate the sharing of photos, videos, and other information on the daily life of the interviewees with their companion dogs. Among the owners, four reinvited me to accompany them on a dog walk after the first walking experience. The interviews with the urban empty nesters mainly covered (but were not limited to) the following six aspects: (1) the motivation for and experience of keeping companion dogs; (2) the relationship with the companion dog (role, attitude, evaluation, etc.); (3) special events in the raising process (medical treatment, travel, etc.); (4) the activity range of the companion dog, the tracking of daily outdoor activities and the interaction during dog walking; (5) personal beliefs and values related to keeping companion dogs; and (6) the owner’s lifestyle (life and changes, etc.). Moreover, during the interview, the owner was encouraged to recall and reflect as much as possible on his or her impressions of the companion-dog-raising experience to explore as comprehensively as possible the multispecies families created jointly by empty nesters and their companion dogs, the leisure interaction between them, and the potential effect on the old owners’ health. In the empirical section, I will present some daily accounts of empty nesters and their companion dogs to reveal how they form multispecies families and participate in leisure activities together. Additionally, some of the views and comments of the empty nesters on the public space that is accessible when walking dogs are provided to discuss the inclusiveness of public leisure space during urban development.

## 3. Results

As the term “empty nest” implies, empty nesters lack support from family members. Therefore, creatively building strong social support for empty nesters is vital to the quality of their later life. For empty nesters, one of the positive ways to thrive in old age is to live and engage in leisure activities with their companion dogs; living and engaging in leisure activities with companion dogs have been proved in our study to be effective in channeling more support to seniors through a wider social network. 

### 3.1. Multispecies Kinship and the Construction of Companion Dog-Owning Life in Old Age

#### 3.1.1. Meanings of Multispecies Family and Emotional Connectedness

Companion dogs as animal actors can help empty nesters rebuild their later life, and together, both groups jointly create a new meaning for the multispecies family through companionship, thereby forming a multispecies kinship. This kinship is exemplified by typical events, such as the experience described by interviewee No. 4. *Cookie* (a 3-year-old Standard Poodle) is a companion dog who was given by interviewee No. 4 to her 87-year-old mother. The mother is a typical empty nester, as she is living alone because her daughter already has a core family. Over time, the mother became disabled; she was unwilling to communicate with others and was reluctant to go outside. Over time, her physical condition deteriorated due to a lack of exercise until she could not even go out. Inspired by her friend’s suggestion that she give her mother a puppy, interviewee No. 4 bought one for her mother and asked her to take care of it. Now, *Cookie* lives with the older mother. As a companion dog, the dog fills the void left by interviewee No. 4 and brings new vitality to her mother’s later life. This emotion transcends the joy brought by a pet in the usual sense. *Cookie* has, to a large extent, helped the older lady regain the meaning in her life by giving her non-human support that her daughter could not provide.

In addition to rebuilding their owners’ lives, companion dogs facilitate an emotional connection between members of the original family. As a single mother who became pregnant late in life, interviewee No. 2 raised her daughter by herself. Her daughter mentioned the loneliness of being the only child more than once, and later, the daughter even experienced minor depression due to academic pressure. To ease her symptoms, the mother agreed to her daughter’s proposal to obtain a puppy, *Niuniu*, a 2-year-old sheepdog. Her daughter considers herself to be the “mother” of *Niuniu* and the interviewee to be the “grandmother”. The arrival of *Niuniu* enabled the divorced interviewee No. 2 and her daughter to form a loving same-sex multispecies family. She has been living with *Niuniu* since her daughter left home to study abroad.


*“We thought of her (Niuniu) as a family member. My daughter said, although she had raised so many little animals before, she had never felt that way. Niuniu is just like part of our family, and it is also a comfort to me. My daughter is studying abroad. I often take videos and share funny stories about Niuniu with her. Because of the dog, we feel closer to each other. What’s more, I don’t feel lonely though I’m living alone now.”*
*——Interviewee No. 2*

The support, especially the emotional support, provided by pets is explicitly recognized by most pet-owning families [36]. This recognition was also clearly shown in the interview with interviewee No. 2; in this interview, the role of companion dogs in empty-nest households was highlighted as the interviewee talked about the solace she finds in such an intimate relationship and how her daughter sees herself as the dog’s parent. Without effective support from family members, empty nesters cannot share their daily lives with others; not sharing their daily lives with others would make the empty nesters feel lonely or even depressed [37]. Fortunately, *Niuniu* has been a part of the family for No. 2, and the dog is regarded as a sentient being rather than another animal. In taking care of *Niuniu*, No. 2’s family has forged close emotional bonds and obtained intimate non-human support.

#### 3.1.2. Agency of Both Companion Dogs and Urban Empty Nesters

Companion dogs not only can provide support as family members but also can play an active role in the seniors’ later life by exercising agency. The following is a good case in point. Before retirement, interviewee No. 3 worked at a university. After years of living at a busy pace, she worried that doing nothing would be bad for her health. To keep herself occupied, she participated in a variety of activities organized by the community club and adopted *Domi*, a 5-year-old Miniature Poodle, as a companion. Below, the interviewee describes one of her happiest moments in raising a puppy:


*“Every day when you go back home, Domi will always be the most enthusiastic one in our family, even more than my husband. He jumps on you, up and down, with sparkling eyes. I even felt he (Domi) was out of control. Then I thought, wow, am I that popular? I feel a strong sense of existence, and then I’ll touch Domi right away.”*
*——Interviewee No. 3*

*Domi* expresses affection and expectation through body language, such as touching and begging for petting, thereby making interviewee No. 3 feel that she is needed. Touching has been proven to be mutually beneficial, both physically and mentally, for owners and their pets [38]. Stebbins [39,40] powerfully supports the view that sensory stimuli, such as touch, are a leisure interaction. Additionally, interviewee No. 3 mentioned that she was fond of buying clothes for *Domi*, especially in winter, when she would buy several down jackets. In her opinion, *Domi* needs to put on more clothes, just as a human would, when the weather gets cold; this opinion illustrates the difference between *Domi*, as a non-human family member, and other animals.


*“I think Domi is to me what a baby is to Mommy. My son doesn’t come to see me often. He’s very busy.”*
*——Interviewee No. 3*

Most researchers hold that strong social support is good for seniors, particularly for empty nesters [41,42,43]. From a psychological perspective, owners tend to regard companion dogs as “little humans” rather than “things” [44]. We thus infer that non-human animals are crucial to the multispecies kinship of empty-nest families, in which humans cease to be the center and the boundaries between species are indistinct [45]. Such a relationship represents intimacy and a sense of belonging and transcends species to become the most important and enduring bond in urban empty nesters’ everyday lives. This relationship is by no means a one-way process, thus indicating that the companion dog is integrated into the family and that the companion dog has been given the same significance as human relatives; the companion dog is like “a family member who can’t talk”, said family No. 1. The research on how empty nesters and companion dogs support each other and share life may explain why pets are, to some extent, better than humans at providing support and companionship to seniors. For urban empty-nest owners, companion dogs not only bring them happiness in a way that other species cannot but also provide unconditional love and companionship, which constitute an important form of social (non-interpersonal) support.

### 3.2. Shared Outdoor Leisure Space: Potential Benefits to Health

Animal welfare is discussed from an anthropocentric perspective emphasizing “beastly nature” [46]. The outdoors usually serves as a perfect place for dogs to unleash their beastly nature and to participate in leisure activities with their owners and is also a space for urban empty nesters to participate in leisure activities with their companion dogs. On the one hand, owners personify companion dogs’ agency [47], i.e., the owners recognize that the right to take leisure belongs to humans and animals. On the other hand, owners believe it is in animals’ nature to go outdoors [48]. Outdoor activities can both satisfy the needs of companion dogs and increase exercise for urban empty nesters, thereby making going outdoors an important practice for the development of multispecies kinship and maintenance of non-human support. 

#### 3.2.1. Rhythm and Co-Discipline

In outdoor activities, urban empty nesters and companion dogs share the same rituals and rhythms and motivate each other until they reach a state of self-discipline. An example is provided by family No. 1 and its dog, *Listen*, a 7-year-old Miniature Poodle. Every day, the couple takes *Listen* with them outdoors to the playground for exercise, whether it is rainy or sunny. However, they have to tie *Listen* to the fence of the playground because the playground is off-limits to dogs. The wife likes jogging, while the husband prefers to walk in circles. Each time the husband comes near *Listen*, the husband claps his hands in an “xx-xx” rhythm in response to *Listen*’s waiting (Figure 1). “*This is our tacit interaction*”, said the husband. He does this is to tell the dog that it has not been abandoned (*Listen* was abandoned by his former owner) and to show the dog appreciation for keeping them company. For the couple and Listen, the daily outdoor exercise is a leisure activity they do as a family, and the activity enables them to build an intimate relationship. Although *Listen* does not directly participate in the couple’s exercise, the perceptions and positive mood created during that period lay the foundation for the co-discipline of the family. The story of *Bubu* and its owner also provides insight into co-discipline. Interviewee No. 7 is an empty nester who lives alone. Like couple No. 1, he goes for a walk with *Bubu*, a 4-year-old Miniature Poodle, after sunset. He spends most of the time outdoors talking to *Bubu* on a chair while holding the dog in his arms as if *Bubu* were a baby. For the empty nester, holding the dog this way is an effective way to communicate with his companion dog. Although it is hard to define whether it is the owner keeping his dog company or the other way around, their time outdoors is undoubtedly extended due to co-discipline, execution, fixed leisure time, and shared ritual. The sharing of key rituals, as Sanders [49] argues, is a major source of cohesion between owners and pets.

#### 3.2.2. Happiness and Mental Health

Kaneko et al. [50] suggest that the social support network is an important factor in ensuring the happiness of empty nesters in their later years. Lack of social and physical activity among older people results in poorer quality of life and, thus, affects the well-being of seniors. Our research suggests that the happiness of empty nesters increases and that their mental state improves when they engage in outdoor leisure activities with their companion dogs because doing so not only makes the empty nesters feel better but also gives them a stronger sense of support. For example, interviewee No. 8 rarely had the opportunity to socialize because she spent most of her retired life taking care of her husband, who was in poor health, and doing household chores. In her fourth year of retirement, she adopted *Didi*, an 8-year-old Medium Poodle. While walking the dog in the neighborhood, she met some friends doing a square dance and joined them. It did not take long before *Didi* became popular among members of the dancing club. Now, her daily routine is to go dancing with *Didi* in the morning and take the dog for a walk at night. When she goes dancing, *Didi* stands by her side and waits (Figure 2). This routine serves as good evidence for the opinion expressed by Glass et al. [51] that social activities are positively related to the mental health of the senior. In the case of interviewee No. 8, *Didi* not only gives her non-human support but also helps her obtain interpersonal support through an extended social network.


*“My son and daughter-in-law do not live in Guangzhou. I raised Didi to kill time. Thanks to this, I go out every morning and night to exercise.”*
*——Interviewee No. 8*

A dog groomer grew fond of *Didi* because of the dog’s standard profile. Thanks to the groomer, *Didi* won *Best Look* in a national pet show. Interviewee No. 8 takes great pride in *Didi*’s experience, and she is, therefore, more willing to take *Didi* out. Additionally, interviewee No. 10 shares the same feeling because her dog, *Kaka*, a 2-year-old Old English Sheepdog, receives many compliments from other pet owners for its gentle temperament. The existence of vicarious pleasure suggests that dog walking brings happiness [47] and increases the likelihood of taking exercise, and these two benefits can be mutually reinforcing.

#### 3.2.3. Outdoor Exercise and Physical Health

Older people with impaired self-care ability will suffer from a decline in physical flexibility, social adaptability, and confidence due to a lack of exercise [52]. Exercise, including dog walking, is important for empty nesters, especially those with limited self-care ability because exercise can effectively slow decline and improve physical health as their bodies age. Such a viewpoint is exemplified by the case of *Cookie*’s granny, one of our interviewees and a typical advanced-age empty nester. Though she has never enjoyed exercise, she goes out twice a day to walk with her companion dog. She thinks walking is not for herself but for her dog. Fortunately, she has recovered much of her self-care ability through dog walking, which increases her time outdoors and engages her in exercise.


*“My mother would stay at home all day without Cookie. This dog is for her so that she will have a sense of responsibility. If my mother doesn’t go out, the dog won’t be outdoors on time. So, she has to go. She always says she doesn’t want to go out and she is too tired to move. Keeping a companion dog is good for my mother. Now she’s much better than before.”*
*——Interviewee No.4*


*“I like Cookie. I go out every day, as long as Cookie wants to.”*
*——Cookie’s granny*

Exercise capacity declines with age. According to relevant studies, 26.9% of the population aged between 65 and 74 and 35.3% of those aged above 75 experience energy loss [53]. Physical activity is crucial to healthy aging; for example, the World Health Organization has listed disease caused by physical inactivity as the fourth-leading global risk for mortality [54]. One of the practical and long-term measures older owners can take to restore physical activity is to be responsible caretakers and walk their companion dogs, just as *Cookie*’s granny does. The three cases above prove companion dogs’ ability to motivate owners to overcome mental and physical challenges [55]. For empty nesters, keeping companion dogs provides an opportunity for them to take care of others. Specifically, the elderly must do their best to care for the health of their companion dogs, which also need attention. Thus, meaningful activities, such as feeding, soothing, and walking are performed; these activities, in turn, benefits owners’ health both mentally and physically.

### 3.3. Negotiation in Outdoor Leisure Interaction: Socio-Spatial Routines and Normal Animal “Citizens”

In fact, with companion dogs gradually entering urban families, Guangzhou Kennel Management Regulations were promulgated on 27 November 2008, to promote the standardization and institutionalization of dog raising. Article 24 strictly prohibited dogs from entering certain public places, such as government agencies, schools, hospitals, children’s places, art galleries, restaurants, transportation hubs, and scenic spots [56]. Therefore, the scope of dog walking in the public space of the city is limited. This exclusion of pet dogs from certain public places is different from the inclusion of pet dogs advocated by Western society and reflects the binary separation between human and non-human space in urban development in China. With the rapid increase in pet dogs, there is an apparent conflict between the demands of people with dogs for diversified lifestyles and the demands of people without dogs for personal safety and clean living environments; this conflict leads to increasingly intense competition between dogs and citizens for urban public space.

#### 3.3.1. Scarcity of Age- and Dog-Friendly Space for Leisure 

The interactive outdoor activities pursued by empty nesters and their companion dogs are not day-round activities without space limitations. Despite the growing tolerance and acceptance of domestic pets in most Chinese cities, dogs are still different from “humans”, after all, in that they are only admitted into some public spaces. In the newly revised *Guangzhou Kennel Regulations*, there are strict rules on the breed and height of pet dogs (no higher than 71 cm), and 22 kinds of ferocious dogs are banned, including Mastiffs, Bulldogs, sheepdogs, and hounds. Even breeds popular in the West are not allowed; these breeds include Staffordshire Bull Terriers, German Boxers, and Rottweilers. The regulations also include updated access restrictions, expanding from public areas with clear boundaries to include roads, bridges, pedestrian overpasses, and underground passages, thereby further shrinking the activity space for dogs. On the other hand, seniors ambulate mainly in their neighborhoods as their body functions deteriorate over time, and the leisure path the older people choose may conflict with the rules and norms of city management. Therefore, older people still face multiple negotiations regarding how to find a proper space for companion dogs in the norms of urban management without affecting the normal life of non-keepers of companion dogs.

Urban empty nesters indeed have much more time to live with their dogs, but empty nesters are cautious in the decisions they make about their socio-spatial walking routines. *Niuniu*’s story, for instance, fully illustrates the point. *Niuniu* is forbidden in any public space and never leaves its private home in the H community except to visit the pet hospital, even though the dog is registered at the local authority for dog management. Interviewee No. 2 indicates that *Niuniu*’s routine outdoor walk usually occurs after 11:00 p.m. to avoid disturbing pedestrians and vehicles and to prevent unnecessary conflicts (Figure 3).


*“Dogs are forbidden in our community, especially such big ones. There is no place to go during the daytime. We can only walk around at night when the roads are less crowded. We try to go where there are no people. Niuniu can only lie at the window to look out during the daytime. The poor girl.”*
*——Interviewee No. 2*

*Niuniu*’s granny mentioned that she worked for many years and rarely traveled. Because she wanted *Niuniu* to enjoy leisure time as she does, she found a pet tour after making inquiries. The person in charge of the tour chartered the hotel and restaurants, and there was a special coach to transport the owners and their dogs. The owners just needed to pay a deposit for check-in. Huizhou was the first destination of *Niuniu*, which is a city far away from Guangzhou. They stayed at a pet-friendly bed and breakfast with the backyard leading directly to the beach. Granny found that *Niuniu* was quite happy and enjoyed herself, so she arranged another trip half a month later (Figure 4).


*“Huizhou boasts wonderful beaches. I put on shorts and went to the sea. She (Niuniu) has been swimming around me happy. Unlike at home or even in ordinary times outside, she ran at a brisk pace on the beach, filled with joy. You know that medium and large dogs have no public place for activity in big cities.”*
*——Interviewee No. 2*

Due to the high price of pet tours and the time commitment, granny is not strong enough to take her dog on long-distance trips. The night walk is a viable option for *Niuniu* to at least commune with its beastly nature and guarantees the dog’s basic “human” rights as a family member.


*“It will be a social problem over time. There are parks dedicated to dogs abroad. I’m envious.”*
*——Interviewee No. 4*

In another talk with interviewee No. 9, a care worker, she mentioned walking the employer’s dog *Xiaohei*, a 5-year-old Labrador. The dog’s inclusion on the new restriction list makes the work even harder:


*“Afraid of being complained of by someone, we pay special attention to the time and place of going out. The dog will be confiscated upon any report. We have no other option but to be careful.”*
*——Interviewee No. 9*

In the revised regulations, the government solicits input from caretakers and plans to define a specific zone in appropriate parks and residential quarters and to set up warning signs to facilitate dog walking during specified time slots. The draft regulation is still open to public suggestions, and building dog parks require careful consideration. The scarcity of friendly human-animal interaction spaces in the planning of liveable communities poses a tremendous challenge for empty nesters seeking to walk their companion dogs nearby.

#### 3.3.2. Responsible Owner and Standardized Non-Human “Citizenship”

In addition to avoiding the regular travel hours of citizens in a limited space, the older owner must develop responsible dog-raising attitudes and help their companion dogs become normal non-human “citizens” and gain access to urban survival permits. The revision of regulations governing dogs in public spaces suggests the dire need for institutional improvement, as the prior regulations failed to accommodate urban development. For civilized dog raising, cleaning dog feces is a top challenge for urban dog management and a focal point of public concern, as seen in solicitations and complaints. When I participated in dog walking with interviewee No. 5, her companion dog, a 13-year-old Bichon Frise named *Toffee*, defecated on its accustomed brushwood. The owner quickly removed the feces and said,


*“I won’t let her just make a mess on the grass. If the social environment deteriorates due to irresponsible dog keepers, dogs won’t be welcomed or tolerated. Dog owners should be careful. Many parents remind children to stay away from the lawn because there may be dog feces. We have to carry a dog poop pick-up bag with us.”*
*——Interviewee No. 5*

Defecation in a highly urbanized public space goes against the norms of a human-centered society and is considered a non-human, inappropriate behavior. To encourage active registration and good dog-walking habits, the dog management authority has been offering a gift package that includes a leash, a pooper scooper, a sign, and a dog mask. If the authority detects any behavior that impacts public sanitation and security, owners will receive a fine, or their dogs will be confiscated. In addition, the government is planning a robust public monitoring mechanism that allows citizens to report any irresponsible dog-raising behavior. Believing that public recognition is the basis of a better environment for dogs, interviewee No. 5 pays close attention to the etiquette of *Toffee* outdoors and tries to civilize the dog’s beastly nature, thereby creating heightened responsibility for herself in the public space. Both interviewees No. 5 and No. 1 desired a dog park to be marked out in their communities because such a space can help release the nature of the dogs and save their energy. A dog park would also create a welcoming space for engaging in outdoor activities and maintaining an intimate relationship with companion dogs.

As seniors pursue a positive later life through multispecies leisure activities, their choice of socio-spatial routines and a series of responsible raising behaviors demonstrate their longing for hospitable public space. China has been improving animal welfare through legislation in recent years, but the focus remains on the meat food trade, thus neglecting animal care. Dog-related regulations emphasize city management and citizen rights more than dog-raising etiquette, which is mentioned only in slogans. We hold that reasonable and dog-friendly regulations strictly safeguard the rights of both humans and animals. There must be a humanistic approach to dog raising in China to institutionally secure the basic rights of dogs, such as the right to participate in outdoor activities, and realize the harmonious coexistence of owners, non-caretakers, and dogs. Considering the narrower living space of the elderly [57], we call for welcoming public spaces for both senior citizens and animals because providing such spaces demonstrates care for the aged (especially empty nesters and companion dogs) in the context of rapid population aging and urbanization and has positive implications for the building of future sustainable cities.

## 4. Discussion

Empty nesters, as a new group, have posed a remarkable challenge to China’s coping with an aging population amidst rapid urbanization. Finding a creative way to enrich the social support network of urban empty nesters and improve their quality of life would offer an effective solution for them to actively cope with aging. This paper provides a grounded understanding of how companion dogs provide non-human support for urban empty nesters.

This paper discusses how companion dogs are involved in multispecies kinship in urban empty-nester households in China and explores the dilemmas and negotiations of urban empty nesters in their daily interactions with their companion dogs from a multispecies leisure perspective. As “companions and family members” who cannot speak, animals have indeed made considerable contributions to enriching humans’ leisure experiences. As Buller [58] said, “the animal presence and influence are finally felt by human beings”, and it is important to focus on the daily interaction between human and non-human animals from the perspective of multispecies leisure in the context of the increasingly close relationship between urban empty nesters and companion dogs in China. We argue that companion dogs can reshape the daily lives of urban empty nesters and create a new family structure. Keeping companion dogs helps seniors regain intimacy and a sense of belonging in their daily life, thereby challenging the care receiver role. Furthermore, we believe that living with companion dogs is a positive lifestyle for urban empty nesters, as doing so can help meet the demand for social support and prevent the passive and negative ageism arising from aging and identification as an “aged citizen”. In other words, social support for empty nesters is not limited to leisure practices between human subjects but also includes multispecies interactions between humans and animals in an embodied space. We pay attention to the dynamics of companion dogs integrating into the everyday life of urban empty nesters, thereby exploring multispecies interaction.

In contrast to previous studies on the attachment between the older people and pets [59,60] and the connection between therapeutic dogs and old patients [61,62], we have incorporated the interaction between the old people and companion dogs into the urban space while reflecting on the rapid urbanization in developing countries, the inclusiveness of public space and the acceptance of animals as citizens. The public space in post-reform China is vastly different from that in the West. As a miniature of the country’s traditional cultural logic of collectivism, public space has gradually evolved into the site of collective social life for inhabitants by, e.g., hosting square dancing, Tai Chi exercises, and Chinese chess [63], and is dominated by older people and organized activity groups [64]. However, excessive attention is given to collective leisure activities, and the perceptions of these activities as being collective inevitably lead to the scarcity and even absence of public spaces for seniors and living spaces for animals.

China’s urban policies are primarily state institution directed and economic growth oriented [65] and do not consider a differentiated population structure and “non-human citizens”, thus posing unprecedented challenges for liveability and sustainability. These challenges stem mainly from city management problems (such as a lack of dog registration, difficulties controlling stray dogs, dog attacks, and the lack of self-discipline among citizens) and insufficient cooperation with the relevant management department. As a result, in 2019, the Guangzhou local government began revising the decade-old kennel regulation to further institutionalize dog ownership. The amendments focus mainly on the details of the dog registration system, compulsory immunization measures, common governance by social subjects, the legal repercussions of illegal dog keeping, the division of duties among government departments, and law enforcement measures. By July 2019, the number of registered dogs in Guangzhou had grown rapidly to 106,000 [56], thereby leading to the harassment of non-caretakers in public spaces and creating pressing needs for urban management. Therefore, to balance the rights and interests of citizens in daily life and promote urban management, dog keepers must, both morally and financially, assume the responsibility of securing their dog’s limited public space in the urbanization drive. However, friendly spaces, such as dog parks, for senior citizens and their dogs are not incorporated into the urban planning system, and related policies are yet to be designed. We hold that how to provide the public with vibrant, year-round public spaces jointly created by humans and their companion animals should be considered in city construction.

The data of this study were obtained through direct observation by the first author (who participated in the outdoor activities of empty nesters and their companion dogs) rather than through static observation and interviews. The authors reviewed the life stories shared by the empty nesters during the dog walks and reflected on the inclusiveness of urban leisure space according to the outdoor dog-walking route. Therefore, this study is an embodied and interactive qualitative study. Our research, which is based on the agential nature of both humans and animals, provides a useful reference for leisure scholars probing human-place leisure interaction. We hold that animals are not in a completely passive position in the human-centered world but play a dynamic part in the social network woven together by humans and non-human animals. However, the study has some limitations. We chose a community with a high concentration of empty nesters and open activity spaces, where the control of dogs is not strict; thus, this community is conducive to providing some vivid qualitative materials. In addition, it would be valuable to conduct further studies that explore empty nesters with a great deal of heterogeneity, including gender, marital status, living arrangements, and housing or whose living space is not as conducive to the survival of companion dogs as the H community. 

## 5. Conclusions

In conclusion, creative forms of social support, especially the non-interpersonal support and multispecies interaction that this study focuses on, can enrich and expand the social network of urban empty nesters. Raising companion dogs, sharing leisure space, and carrying out outdoor activities regularly with them are positive ways to reconstruct the everyday life of the urban empty nesters. For the prospects of urban multispecies interaction, the possible key point is the extension of friendly public space.

## Figures and Tables

**Figure 1 ijerph-17-04091-f001:**
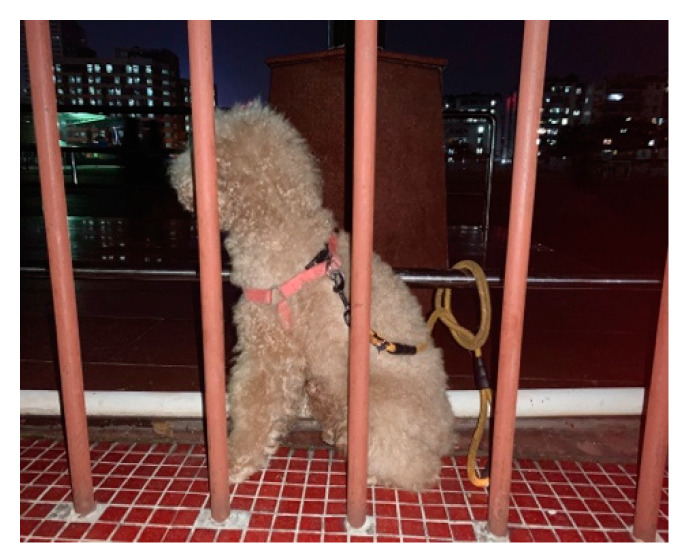
*Listen* is waiting while the couple interviewees No. 1 exercise.

**Figure 2 ijerph-17-04091-f002:**
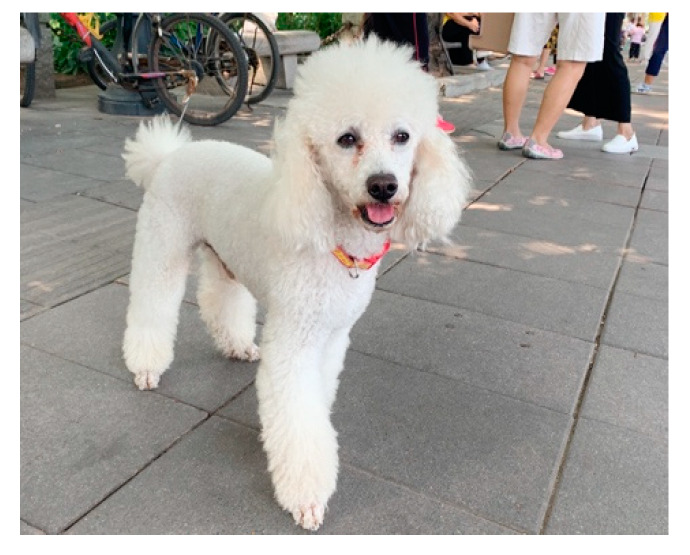
*Didi* accompanies interviewee No. 8 to a dance.

**Figure 3 ijerph-17-04091-f003:**
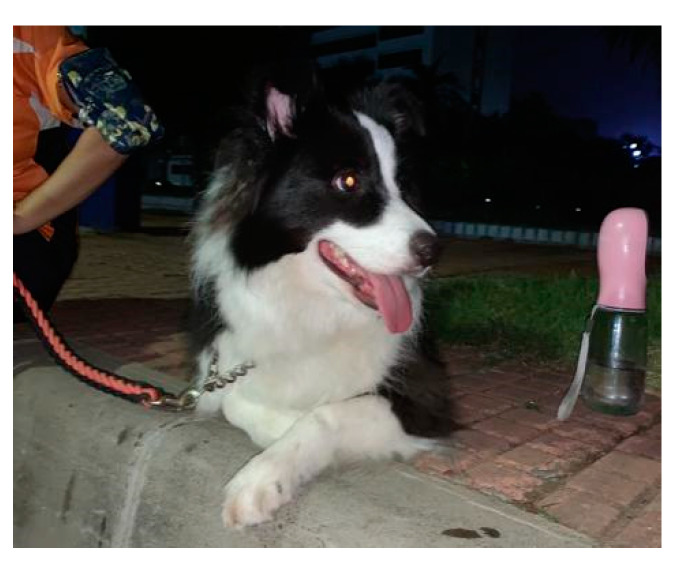
*Niuniu* still walking at midnight.

**Figure 4 ijerph-17-04091-f004:**
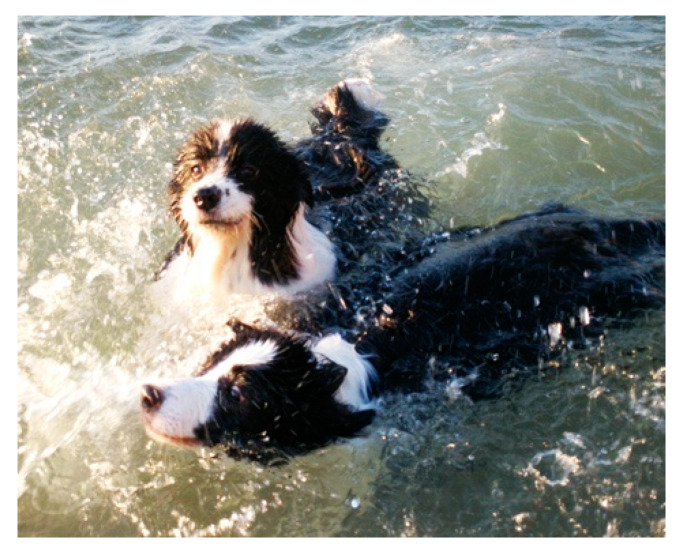
*Niuniu* goes on a pet tour at the seaside and plays with a companion.

**Table 1 ijerph-17-04091-t001:** Interview information on the urban empty nesters and their companion dogs.

Interview	Gender	Age of Urban Empty Nester	Occupation	Name	Type	Size	Age of Companion Dog	Rearing Time	Frequency Walked	Daily Activity
1	Female	61	Freelancer	Listen	Poodle	Miniature	7	4 years	Twice	1.5 h
	Male	64	Freelancer							
2	Female	62	Retired	Niuniu	Border Collie	Medium	2	2 years	Once	1 h
3	Female	65	Retired	Domi	Poodle	Miniature	5	5 years	Once	45 min
4	Female	88	Housewife	Cookie	Poodle	Medium	3	3 years	Twice	1 h
	Female	60	Retired							
5	Female	73	Retired	Toffee	Bichon Frise	Miniature	13	7 years	Once-twice	1 h
6	Female	64	Retired	Yuanyuan	Papillon	Miniature	7	7 years	Twice	1 h
7	Male	71	Retired	Bubu	Poodle	Miniature	4	4 years	Twice	1 h
8	Female	67	Retired	Didi	Poodle	Medium	8	8 years	Twice	2 h
9	Female	61	Caregiver	Xiaohei	Labrador	Large	5	5 years	Twice	1 h
10	Male	65	Retired	Kaka	Old English Sheepdog	Large	2	2 years	Twice	1 h

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
