# Peer review of "Everyday Life Construction, Outdoor Activity and Health Practice among Urban Empty Nesters and Their Companion Dogs in Guangzhou, China"

_ijerph, 2020, doi:10.3390/ijerph17114091_

Round 1

Reviewer 1 Report

Only two suggestions

Consider giving a small background on the shift in attitudes towards companion animals in society. I find it extremely interesting to see the dramatic change in the last decade. You touch on it but consider expanding some. 

Second did the study respondents sign releases and was the study reviewed by a IRB 

One other thing regarding vet care for the companion animals.  Is there a financial hardship for these older adults to provide appropriate veterinary care for animals?  

Do you see possible changes in governmental policy regarding access for companion animals in public spaces?

Author Response

Response to the reviewers

Dear Prof. Tchounwou and Reviewer 1:

Many thanks to your and the reviewer 1’s efforts in assessing our manuscript entitled “Everyday Life Construction, Outdoor Activity and Health Practice among Urban Empty Nesters and Their Companion Dogs in Guangzhou, China”. We deeply appreciate all the constructive comments and advices. We will detail our responses to the reviewers as follows.

Reviewer 1

  1. Consider giving a small background on the shift in attitudes towards companion animals in society. I find it extremely interesting to see the dramatic change in the last decade. You touch on it but consider expanding some.

Much appreciate for reviewer’s constructive suggestion. We expand on the nearly 40-year shift in social attitudes toward companion animals in China and focus on several important points (see Page 2, L75).

  1. Second did the study respondents sign releases and was the study reviewed by an IRB

Many thanks for this constructive suggestion. This study focuses on the understanding and reflection of the relationship between the elderly and their companion dogs, especially focuses more on the empirical perception of urban empty-nesters. There was no ethical conflict with animals in the study, which was not reviewed by the IRB but was reviewed and supported by the academic research committee of the researcher's unit.

We informed respondents of the purpose and method when invited them to involve in our study, signed the informed consent after the respondents' consent. We explained this in the revised methodology (see Page 4, L164).

  1. One other thing regarding vet care for the companion animals. Is there a financial hardship for these older adults to provide appropriate veterinary care for animals?

Many thanks for the reviewer’s constructive comment. All of our respondents had no financial hardship, and some had a retirement pay, while others had a government basic living allowance and received financial support from their children. Care for the empty nester’s dog is limited to vaccinations and insect repelling. They don't need extra care like dressing up,which is more popular for the younger owner in China

We are very grateful to the reviewer for inspiring me with his comments. It would be valuable to further studies to discuss how the poor old owner interact with their companion dogs and negotiate with public space. The diversity of the demographic structure of the subjects referred to the outlook section at the end of the discussion (see Page 13, L545).

  1. Do you see possible changes in governmental policy regarding access for companion animals in public spaces?

Many thanks for the reviewer’s comment. Based on the analysis of dog regulations in major cities of China, we believe that the government's policy regarding access for companion animals in public spaces will not change for the moment, and blind persons carrying guide dogs and severely disabled persons carrying disabled dogs are excluded. But the government may formulate policies to plan special areas for animal outdoor activities.

We hope that our revisions have improved the quality of the paper and we would much appreciate if the Journal could reconsider this submission.

With thanks and best wishes

The Author

Reviewer 2 Report

I have made many comments in the manuscript. I think the paper should be improved by

1- reducing the redundancy, many things are said several times

2- clarifying the specific aim of the study

3- clarifying the originality of the study. Many studies have been realised on the impact of pets on well being of elderly people. What is specific with this study ? 

4- the method is original and should be more thoroughly explained and justify. Please see my other comments in the manuscript itself.
